# Improving Reporter Gene Assay Methodology for Evaluating the Ability of Compounds to Restore P53 Activity

**DOI:** 10.3390/ijms232213867

**Published:** 2022-11-10

**Authors:** Xinle Han, Jun Du, Dandan Shi, Lingjie Li, Dandan Li, Kun Zhang, Suwen Lin, Jingzhong Zhu, Zoufang Huang, You Zhou, Zhengyu Fang

**Affiliations:** 1Biomedical Research Institute, Shenzhen Peking University—The Hong Kong University of Science and Technology Medical Center, Shenzhen 518036, China; 2Department of Pathology, Ninth People’s Hospital, Shanghai Jiao Tong University School of Medicine, Shanghai 200011, China; 3Department of Hematology, Renji Hospital, School of Medicine, Shanghai Jiao Tong University, Shanghai 200127, China; 4Shenzhen Chipscreen Biosciences Co., Ltd., Shenzhen Hi-Tech Industrial Park, Shenzhen 518057, China; 5Ganzhou Key Laboratory of Hematology, Department of Hematology, The First Affiliated Hospital of Gannan Medical University, Ganzhou 341000, China; 6Department of Pathology, Peking University Shenzhen Hospital, Shenzhen 518036, China

**Keywords:** P53, methodology, reporter gene assay, biological activity

## Abstract

Tumor suppressor protein P53 induces cycle arrest and apoptosis by mediating the transcriptional expression of its target genes. Mutations causing conformational abnormalities and post-translational modifications that promote degradation are the main reasons for the loss of P53 function in tumor cells. Reporter gene assays that can scientifically reflect the biological function can help discover the mechanism and therapeutic strategies that restore P53 function. In the reporter gene system of this work, tetracycline-inducible expression of wild-type *P53* was used to provide a fully activated state as a 100% activity reference for the objective measurement of biological function. It was confirmed by RT-qPCR, cell viability assay, immunofluorescence, and Western blot analysis that the above-mentioned reporter gene system could correctly reflect the differences in biological activity between the wild-type and mutants. After that, the system was tentatively used for related mechanism research and compound activity evaluation. Through the tetracycline-induced co-expression of wild-type *P53* and mutant *P53* in exact proportion, it was observed that the response modes of typical transcriptional response elements (TREs) to dominant negative *P53* mutation effect were not exactly the same. Compared to the relative multiple-to-solvent control, the activity percentage relative to the 100% activity reference of wild-type *P53* can better reflect the actual influence of the so-called P53 mutant reactivator. Similarly, relative to the 100% activity reference, it can objectively reflect the biological effects caused by the inhibitor of *P53* negative factors, such as MDM2. In conclusion, this study provides a 100% activity reference and a reliable calculation model for relevant basic research and drug development.

## 1. Introduction

Tumor suppressor protein P53 is a pivotal transcription factor and cellular stress sensor activated by various stresses, such as DNA damage and oncogene activation. By transcriptional up-regulation, P53 can induce cell cycle arrest via the cyclin-dependent kinase (CDK) inhibitor *CDKN1A* (also known as *P21*) [1] and induce apoptosis through the BH3-only proteins *PUMA* and *NOXA* [2,3,4]. Some studies have also found that it plays an essential role in ferroptosis and the regulation of cellular metabolism [5,6]. However, *P53* is the most frequently mutated gene, with a conservative estimate of the overall mutation rate at 34.5% across all tumors [7]. Missense mutation represents the most common variation type in the *P53* gene, with widespread distribution of location sites [8,9]. If P53 loses its original function due to mutations, cells may escape senescence and apoptosis. Furthermore, some mutant P53 proteins could acquire new oncogenic properties called gain of function (GOF) [10]. In other circumstances, such as in acute myelogenous leukemia (AML), the overexpression of *MDM2*, rather than a loss-of-function (LOF) mutation in *P53*, disrupts *P53* function by ubiquitinating the protein and promoting its degradation and inactivation [11,12,13]. The restoration of wild-type function to mutant P53 (mutP53) with a small molecule (hereinafter referred to as “mutP53 reactivator”) is among the current holy grails in cancer therapeutics [14]. Regardless of the strategy for drug discovery, recovery of the biological function of P53 should be the necessary goal while evaluating the target effect. Thus far, screening and evaluation methods of P53 reactivation agents remain controversial.

In a cell-free system, the method based on fluorescence resonance energy transfer (FRET) technology can evaluate the affinity between the DNA binding domain (DBD) of P53 and the promoter fragment of the target gene [15]. However, it has a limited ability to directly reflect changes in the transcriptional activity of P53. Cell viability tests can reflect the cytotoxicity of compounds to tumors. Still, it cannot distinguish between the target effect and off-target effect. At a cellular level, the biological activity of transcription factors can be determined using real-time quantitative PCR (RT-qPCR) and reporter gene assays. However, the former is a cumbersome, time-consuming, and low-throughput operation. Thus, the classical luciferase reporter gene assay, which can not only reflect biological effects but also have high detection throughput, is expected to be used for screening and early evaluation of mutP53 reactivators.

As a non-small cell lung cancer cell line that does not express P53 protein, NCIH1299, is usually used as a tool for overexpressing mutant P53 [16,17,18]. In genetically engineered NCIH1299 cell lines that exogenously express different P53 mutants, the effects of compounds could be assessed by comparing the 50% growth inhibition (GI50) values or the fluorescence signal emitted by a simultaneously engineered reporter gene quantitative system. In terms of the reporter gene system, as we all know, inactive P53 mutants could not cause fluorescent signals significantly different from the background. So, after uniformly deducting the background, a fold-change calculation compared to the value close to zero of the solvent group would reflect very limited biological significance. This study focused on improving the existing reporter gene methods and proposing a more suitable algorithm to evaluate compounds that restore P53 activity and provide methodological references to similar studies.

## 2. Results

### 2.1. Tetracycline-Inducible Expression of Wild-Type P53 and Its Mutants

The essential biological function of P53 is to cause cell cycle inhibition and apoptosis. Therefore, the stable expression of wild-type P53 (wtP53) protein is often fatal to the target cells. However, in the reporter gene system, only the expression of wtP53 can reflect the actual transcriptional activation state of the target gene and provide positive control of 100% activation in data calculation.

To achieve the controllable expression of P53, the coding sequences of wild-type and seven reported hotspot mutant P53 proteins were cloned downstream of the Tet-on promoter of pLVX–TetOne-Puro plasmid by double enzyme digestion and ligation so that P53 could be induced by tetracycline antibiotic. The constructed plasmids were packaged into lentivirus and infected NCI-H1299 cells. The engineering cell lines with puromycin resistance were obtained through antibiotic screening. RT-qPCR results show that the expression of *P53* mRNA was significantly induced by DOX for 24 h compared to the cell line infected with negative control lentivirus (NCIH1299-Vector) (Figure 1A). When taking wtP53 mRNA in NCIH1299-wtP53 cell line as a 100% reference, the expression efficiency of mutants in NCIH1299-mutP53 cell lines was 43.3~236.6%, respectively. Except for NCIH1299-mutP53 (Y220C), the expression efficiencies of other mutants were more than 100% that of wild-type expression strains. Western blotting also showed that wild-type or mutant P53 proteins were only expressed in each cell line induced by DOX and the trend of protein expression efficiency was similar to that of mRNA expression efficiency (Figure 1B,C).

Induced by DOX, only NCIH1299-wtP53 can be stained by PAb1620 antibody, while NCIH1299-mutP53 (R273H) and NCIH1299-mutP53 (R282W) can only be stained by PAb240 antibody. After each engineering cell line was treated with an induced dose of DOX for 120 h, the cell viability was detected by alarmarBlue assay (Figure 1D). Compared to the group without DOX, the 7 cell lines expressing mutant P53 protein were similar to NCIH1299-Vector, and there was no significant inhibition of cell viability. NCIH1299-wtP53 expressing wtP53 protein was inhibited by more than 50% under the action of DOX. Different fluorescein-labeled antibodies were used to specifically recognize wtP53 and mutP53 proteins and immunofluorescence analysis was carried out to identify the conformation of P53 protein (Figure 2). The results show that the wtP53 protein induced by DOX had the correct conformation and biological function, while the mutant P53 protein was generated but with no correct conformation and biological function.

### 2.2. Improved Reporter Gene System can Correctly Reflect the Transcriptional Activity of P53

Reporter gene assay is a research method used to detect the interaction between a transcription factor and transcriptional response element (TRE) of the target gene promoter. The expression level of the reporter gene can reflect the transcriptional regulation on the target gene by transcription factor. pGL4.10-PUMA-Luc was transfected into NCIH1299-wtP53 cells, and the expression of luciferase reporter gene downstream of the P53-TRE of *PUMA* was measured with luminescent assay (Figure 3A). The results show that, in the presence of DOX, the TRE of *PUMA* was activated and the luciferase expression level was up-regulated. PFT-α is a small molecule compound that inhibits P53 transcriptional activity. When PFT-α was added into the above-mentioned reporter gene system, the luminescence was inhibited by PFT-α dose-dependent affection, confirming that DOX-induced wtP53 activated the TRE of *PUMA* through its transcriptional activity and the luminescent level of luciferase can directly reflect the transcriptional level of target gene.

Furthermore, pGL4.10-PUMA-Luc was transfected into NCIH1299-Vector, NCIH1299-wtP53, or NCI-H1299-mutP53 cells, and the luciferase activity (relative light unit, RLU) under DOX was detected (Figure 3B). The RLU of NCIH1299-Vector (as a negative control) was 0.55, NCIH1299-wtP53 was 241.72, and cell lines with seven kinds of mutP53 were 0.52~2.42. The results show that, compared to wtP53, the PUMA transcriptional activity retained by the seven P53 mutants was feeble.

RT-qPCR is considered the gold standard for detecting gene expression at the transcriptional level. pGL4.10-PUMA-Luc, pGL4.10-CDKN1A-Luc, pGL4.10-MDM2-Luc, and pGL4.10-BAX-Luc were transfected into NCIH1299-Vector, NCIH1299-wtP53, NCI-H1299-mutP53 (G245S), and NCI-H1299-mutP53 (R282W) cells, respectively. After DOX treatment, reporter gene and RT-qPCR tests were simultaneously carried out. In the reporter gene test (Figure 3C), taking different P53 statuses as groups, RLU values of non-DOX-treated samples were deducted as the background, and NCIH1299-wtP53 was used as a 100% active positive reference to calculate the relative activity of different reporter genes in each cell. Consistent with the *PUMA* reporter gene system, the *CDKN1A*, *MDM2*, and *BAX* reporter gene systems showed that mutP53 (G245S) and mutP53 (R282W) had minimal transcriptional activity on the corresponding genes. In the RT-qPCR test, NCIH1299-Vector was used as a reference to calculate the relative expression of related genes (Figure 3D). wtP53 up-regulated the transcription of four genes by 3.09~14.73 times and mutP53 (G245S) had no significant effect on the transcription of four genes, mutP53 (R282W) had no significant impact on the transcription of three genes, and only mutP53 (R282W) increased the transcription of MDM2 by 2.52 times. Mainly, the reporter gene system with Tet-on-inducible P53 expression can reliably reflect the transcriptional activity of wtP53 and the inactivation of mutP53.

### 2.3. Different Response Modes of Typical Transcriptional Response Elements to Dominant Negative P53 Mutation Effect Have Been Observed

Studies have shown that P53 exerts transcriptional activity by binding with the TRE of the target gene in the form of the homologous tetramer, which is assembled by two pre-assembled dimers [19,20]. When the heterozygous inactivation point mutation of P53 allele occurs, the mutant and wild-type P53 form into a “non-functional” tetramer through the so-called “dominant negative mutation effect (DNE)” [21]. However, it is not completely clear which tetramers are “non-functional” forms and whether heterozygous tetramers have different activities for TRE of varying target genes.

The combination of wild-type and an inactivated point mutation P53 may assemble into three forms of dimers (i.e., wt/wt, wt/mut, and mut/mut), while the three above-mentioned forms may be combined into five kinds of tetramers (i.e., the proportion of wt to mut is 4:0, 3:1, 2:2, 1:3, and 0:4, respectively). Respectively, the wild-type P53 expression plasmid was accurately pre-mixed with different mutant P53 expression plasmids according to the five above-mentioned proportions and transiently co-transfected with different reporter gene plasmids into NCIH1299 cells. When DOX was added, it could be considered that the wild-type and mutant P53 were expressed according to the corresponding specific proportion.

The RLU value of each combination was detected (Figure 4A–G). Grouping by mutation sites and reporter genes, after deducting the background value, the relative activity of the reporter gene in each group was calculated with the group of wt to mut = 4:0 as the 100% activity control. Taking the proportion of wt to mut as the abscissa, the relative activity data of each group were curve-fitted. According to Figure 4A–G, it is not difficult to find that different mutants have roughly the same effects on the transcriptional activity of wild-type P53 but they seem to have different response modes for TREs of various target genes. When the proportion is low, the seven mutants have a potent inhibition on the transcription of *BAX* but have little influence on the transcription of *PUMA*, while the impact on the transcription of *MDM2* and *CDKN1A* is between the above-mentioned two.

Suppose the dimer and tetramer assembly processes are both random. In that case, the probability of forming different dimers can be calculated by monomer proportion and the probability of forming different tetramers can be further deduced (Table 1). There may be four TRE response modes. That is, tetramer can be active only when it contains four (wt = 4), more than three (wt ≥ 3), more than two (wt ≥ 2), or no less than one (wt ≥ 1) wtP53 monomer. According to the probabilities of different tetramers inferred from different wt to mut proportions, the total activity that different response modes may retain can be calculated (Table 1). For example, in the manner of wt ≥ N, relative to a 100% positive control, the total activity that may be retained is the sum of the probability that the tetramer contains N or more wtP53 monomers. When taking the proportion of wt to mut as abscissa, four model curves can be obtained by fitting each mode’s deduced activation percentage (Figure 4H).

By analogy, the P53-TRE of *PUMA* responds to the DNE of P53 with a mode closer to wt ≥ 1. If more than one wtP53 is in the tetramer, the transcriptional activity for *PUMA* can be retained. The P53-TRE of *BAX* responds to the DNE of P53 with a mode closer to wt = 4. That is, the tetramer that can initiate *BAX* transcription must be entirely formed by wtP53. Nevertheless, the response modes of *CDKN1A* and *MDM2* to wt/mutP53 hetero-tetramer are similar to wt ≥ 2 or 3.

### 2.4. Improved Reporter Gene System Could Be Applied to Evaluate mutP53 Reactivators

Previous literature reports showed that COTI-2 and ATO are possible mutP53 reactivators [22,23,24]. In NCIH1299-mutP53 (Y220C), NCI-H1299-mutP53 (G245S), and NCI-H1299-mutP53 (R282W), the reactivation activities of the two above-mentioned compounds on the three mutP53 proteins were tested through the pGL4.10-PUMA-Luc-encoded reporter gene (Figure 5A). Different from previous reports, due to the controllable induced expression of wtP53 and mutP53, the activities of compounds to reactivate mutP53 can be evaluated with NCI-H1299-wtP53 as a 100% activity reference.

It was observed that the expression of wtP53 significantly induced the activity of the reporter gene but, relatively, the treatment of two compounds almost did not enhance the activity of the reporter gene in each mutP53 group. The most potent induction occurred only in the sample of ATO−treated mutP53 (R282W), and the reporter gene activity increased from about 0.09% to about 4.77%. Correspondingly, RT−qPCR showed that the activity of wtP53 increased the expression of the *PUMA* gene by about 2.53 times, while each mutP53 treated with compounds did not cause apparent up-regulation of the mRNA level of *PUMA*; the highest up-regulation occurred in the ATO-treated mutP53 (Y220C) sample, about 1.41 times (Figure 5B). The reporter gene results of COTI-2 and ATO acting on the seven hotspot mutations were shown in Appendix A.

In the presence of DOX, seven kinds of NCIH1299-mutP53 cells or NCIH1299−Vector cells were treated with gradient doses of COTI−2 or ATO. A cell viability test showed that, although COTI−2 or ATO can inhibit cell proliferation in a dose−dependent manner, there was no significant difference in the inhibition curve between cells with different *P53* statuses. The sensitivity of NCIH1299 to so−called reactivators does not depend on the expression of mutP53 (Figure 5C,D).

### 2.5. MDM2 Inhibitors could Be Evaluated Using the Improved Reporter Gene System

Another loss-of-function mode of P53 is that it becomes easy to be degraded due to post-translational modifications, such as ubiquitination, phosphorylation, etc. Although originally transcriptionally expressed by *P53*, *MDM2* encodes an E3 ubiquitin ligase, which promotes the proteasome-dependent degradation of P53 by ubiquitination modification 10. Overexpression or amplification of the *MDM2* gene is frequently detected in various tumors and is mutually exclusive with P53 mutation [25,26]. Blocking the interaction between MDM2 and P53 is one of the critical strategies for restoring P53 function [27]. MI773 and RG7112 have proven to be effective *MDM2* inhibitors [28,29]. Other than cell−free enzymatic assays, the test scheme based on the reporter gene system helps observe the changes in P53 biological function more conveniently after the treatment of compounds.

In our protocol, DOX−treated NCIH1299−wtP53 was used as a 100% activity reference and untreated NCIH1299-wtP53 was used as a 0% activity reference. When *MDM2* was expressed by transient transfection, the activity of *CDKN1A* reporter gene was reduced in a dose-dependent gradient plasmid (Figure 6A). RT−qPCR results also show that the expression of *MDM2* could reduce the transcription level of P53 target genes, such as *CDKN1A*, *PUMA*, and *BAX* (Figure 6B). Based on this, the biological activities of MI773 and RG7112 were observed. The decrease in *PUMA* reporter gene activity caused by overexpressed MDM2 can be effectively restored and accumulated in a dose-dependent manner (0.1, 0.3, and 1 μM for MI773 or 1.25, 2.5, and 5 μM for RG7112) (Figure 6C). Similarly, at the mRNA level, the treatment of 1 μM MI773 or 5 μM RG7112 could reverse and up-regulate the expression of P53 target genes (Figure 6D).

Since MDM2 affects the abundance of P53 protein in cells by post−translational modification, a Western blotting experiment was used to verify the results reported above (Figure 7). In NCIH1299−wtP53 cells, the background abundance of MDM2 was not high and it seemed that the transiently transfected overexpression plasmid did not cause a significant increase in the content of this protein. Interestingly, the abundance of P53 protein decreased significantly after the transient transfection of *MDM2* plasmid, while the contents of MDM2 and P53 synchronously increased after treatment with MI773 or RG7112 (Figure 7B,C). Our observations suggest that MDM2 becomes unstable while causing ubiquitination and degradation of P53 protein. On the other hand, the protein abundance of PUMA and CDKN1A, the target genes of P53, was in direct proportion to P53 (Figure 7D,E).

## 3. Discussion

Reporter gene assay is an essential tool for studying transcription factors’ regulatory activity on their target genes, which provides a research strategy for directly reflecting biological activity. Therefore, it is widely used in basic research and applied research. The tumor suppressor gene *P53*, frequently inactivated in tumors, has attracted much attention in medical research and drug development. It has become a popular direction in anti-tumor research to deeply understand the relevant biological mechanisms and develop therapeutic strategies to restore P53 activity.

The transcriptional regulation by P53 on its target genes is the key to its anti-tumor effect. So, reporter gene detection should be one of the methods used to evaluate the biological function of P53. However, although the research and development progress of several so-called mutP53 reactivators has been promoted to the stage of clinical trials, in previous relevant research reports, few reporter gene test data have been used to evaluate the activity of these compounds [16,23,30,31,32,33,34,35]. One of the possible reasons is that cell cycle arrest and apoptosis are the biological effects caused by P53. Cell models with the continuous expression of P53 protein with complete biological activity usually do not stably exist. This makes the reporter gene system lack the reference with the fully activated state of wtP53 and it is difficult to evaluate the degree of mutP53 reactivation. Calculating the so-called up-regulation multiple based on the background activity of the reporter gene in mutP53 system does not necessarily have strict scientific logic. When the actual biological activity of mutP53 approaches zero, the discussion of the relative multiples becomes less meaningful. Therefore, how to use wtP53 as a reference frame is a problem we need to solve.

This study realized the controllable expression of wtP53 and mutP53 through a conditional induction strategy. In this way, wtP53 expressed only in the test process does not show sufficient cytotoxicity and the induced reporter gene activity can be used as a 100% activity reference to evaluate the relative activity of mutP53 (Figure 8).

The RT−qPCR and Western blotting results showed that wtP53 and mutP53 were strictly expressed only in the experimental system added with DOX and the expression abundance of most mutants was comparable to that of wtP53 (Figure 1A–C). Furthermore, by prolonging the treatment time, the expression of wtP53 induced by DOX can inhibit the viability of host cells but each mutant cannot (Figure 1D). The immunofluorescence experiment of conformation discrimination confirmed that there was P53 protein with correct conformation only in the system of wtP53 expression (Figure 2). Interestingly, different distributions of cytoplasm and nucleus were found in wt and mutant P53 protein-transfected cells. In some cases, cytoplasmic P53 can elicit an apoptotic response by localizing to the mitochondria and activating a direct mitochondrial death program [36,37,38,39]. P53 proteins were mainly localized in the nucleus but some were also in the cytoplasm. Regulation of transcription factor activity by post-translational modifications, such as phosphorylation and dephosphorylation, acetylation, ubiquitination, and SUMOylation, may alter transcription factor subcellular localization [40]. The mechanism involved in the translocation of mutant P53 to cytoplasm will be investigated in the future.

When the controllable wtP53 expression coexists with the reporter gene of its target gene, the groups treated with DOX can be detected to have significant luciferase activity and this effect can be inhibited by compounds that can inhibit the transcriptional activity of P53 (Figure 3A). The luciferase activity induced by each mutant was close to that of the vector group, which was much lower than that induced by wtP53 (Figure 3B,C). The results of RT−qPCR also confirmed that only the expression of wtP53 corresponds to the significant up-regulation of the mRNA expression of its target genes (Figure 3D). It seems that the comparison with the effect of wtP53 can adequately reflect the biological state of mutP53. The reason why we chose the system of inducible expression rather than inducible repression (such as Tet−off) is that it may provide more convenience for high−throughput operation [21].

P53 performs its biological function in the form of tetramer further assembled from a pre-assembled dimer [41,42]. In some cases, wtP53 coexists with mutP53 and forms heterozygous tetramer in different proportions. The incorporation of mutP53 can make the tetramer lose transcriptional activity [43,44]. However, the specific association between the proportion of mutP53 and the activity of heterozygous tetramer is not completely clear. In our work, the transient co-transfection with plasmids mixed in exact proportion and induced expression strategy was applied to facilitate the mixed expression of wtP53 and mutP53 in a controllable proportion in the test process (Figure 4A–G). Assuming that the formation of dimers and tetramers is a random combination process, we can calculate the probability of different dimers and tetramers formed by different proportions of protein monomers through mathematical methods. Furthermore, there may be four modes of inactivation, that is, the tetramer loses transcriptional activity when containing more than one, two, three, or four mutant monomers (Table 1). Based on the above-mentioned assumptions, we can simulate four activity curves corresponding to different proportions of mixed plasmids (Figure 4H). The theoretical model and experiment curves had similar trends, although they were not the same. The mathematical model requires more experiments to verify and improve in the future.

Unexpectedly, by testing the reporter gene activity of the four typical target genes when different mutP53 were co-expressed with wtP53 in different proportions, we have obtained four specific patterns of activity curves (Figure 4A–G). It seems that *PUMA* is less affected by the DNE of P53; only tetramers composed entirely of mutants lose transcriptional activity to this gene. Additionally, *BAX* is significantly affected; if more than one mutant is incorporated, the tetramer does not bind to the TRE of BAX. The inactivation modes on *CDKN1A* and *MDM2* are between the above-mentioned two. Transcription responses mediated by transcription factors, such as P53, are complex. To further understand the mechanisms involved, more experiments should be considered to elucidate the binding of wt/mutP53 to specific binding sites on DNA and other transcription-related proteins. We hope to investigate it in our subsequent work.

Since the DNE of P53 has different effects on the transcriptional activities of various target genes, different TRE may not provide the same complete feedback when evaluating mutP53 reactivators through reporter gene assay: in the reporter gene of *PUMA*, perhaps a quarter of the mutants can bring more than 50% transcriptional activity when they return to normal conformation, while in the reporter gene of *BAX*, even if nearly half of the mutants return to regular conformation, the transcriptional activity may be less than 25%.

The reported two so-called mutP53 reactivators were evaluated with the above-mentioned reporter gene. If wtP53 was used as the 100% activity reference, the relative agonistic activities of the two compounds on the *PUMA* reporter gene were not prominent and were mostly very close to the solvent control group without compound treatment (close to 0%), while about 5% of luciferase activity was detected only in the ATO-treated mutP53 (R282W) group (Figure 5A). Correspondingly, RT-qPCR results did not suggest that compound treatment significantly affected the mRNA level of *PUMA* in host cells (Figure 5B). If we change the calculation and use the luciferase activity data of the solvent control group as the denominator to calculate the relative activity multiple of the compound treatment group, the impact of ATO on the *PUMA* reporter gene can be calculated as “up-regulation of about five times” (Appendix A). Similarly, in *CDKN1A* reporter gene, less than 2% of the enzyme activity obtained after ATO treatment can be calculated as “up-regulation of about two times” (Appendix A). The magnification calculation based only on the solvent control group cannot reflect the changes in objective biological functions. By comparing the growth inhibition of the two above-mentioned compounds on blank cells and cells expressing different P53 mutants, it can also be found that the expression of P53 mutants did not change the sensitivity of cells to the two compounds (Figure 5C,D). Therefore, the two above-mentioned so-called mutP53 reactivators do not affect the biological function of the P53 mutants mentioned in this paper.

The effects can also be well-reflected when P53 negative factors, such as overexpressed MDM2, were introduced into the reporter gene system with controllable wtP53 expression (Figure 6A,B). Such a system can be used to evaluate the biological activity of MDM2 inhibitors (Figure 6C,D). Interestingly, in the Western blotting experiment, we found that the constitutively expressed MDM2 protein, not controlled by P53, seems to degrade itself while causing the degradation of P53 protein (Figure 7A–E).

In conclusion, the effect of fully functional P53 should be used as a reference to evaluate the activity recovery of functionally deficient P53. Through the controllable expression of P53, it provides a 100% active reference and a reliable calculation model for relevant basic research and drug development.

## 4. Materials and Methods

### 4.1. Cell Culture

HEK293T cells were cultured in a DMEM medium (ThermoFisher Scientific, Waltham, MA, USA), and NCIH1299 cells were cultured in RPMI-1640 medium (ThermoFisher Scientific, Waltham, MA, USA). All culture media were supplemented with 10% fetal bovine serum (FBS) (ThermoFisher Scientific, Waltham, MA, USA), penicillin (100 IU/mL), and streptomycin (100 μg/mL) (HyClone, Logan, UT, USA). The cells were maintained in an incubator at 37 °C with 5% CO_2_.

### 4.2. Compounds

COTI-2, MI773, and RG7112 were purchased from MCE (MedChemExpress, New jersey, USA), P53 inhibitor Pifithrin-α (PFT-α) was purchased from Selleck (Houston, TX, USA), and arsenic trioxide (ATO) was obtained from Dr. Zoufang Huang, the First Affiliated Hospital of Gannan Medical University. The storage conditions were based on the manufacturers’ instructions. Doxycycline (DOX), which is an antibiotic derived from tetracycline, was purchased from Meilune (Dalian, Shandong, China) and used as an inducer of the Tet-on system.

### 4.3. Reconstruction of Tetracycline-Inducible Overexpression Plasmids and Lentivirus Packaging

P53 hotspot mutation sequences (including R175H, Y220C, G245S, R248W, R249S, R273H, and R282W) were obtained from the cBioPortal database (https://www.cbioportal.org, accessed on 1 July 2022). The backbone vector pLVX-TetOne-Puro (Clontech, Mountain View, CA, USA) plasmid was used to reconstruct a lentiviral vector-containing wild-type or mutant P53 coding sequence under the control of a tetracycline-responsive element promoter with low-background expression in the absence of tetracycline antibiotics (TCs) and solid transcriptional activity in the presence of TCs. After both were digested with EcoRI (New England BioLabs, Ipswich, MA, USA) and BamHI (Takara, Japan), P53 coding sequences and lentiviral vectors were then ligated using T4 DNA ligase (New England BioLabs, Ipswich, MA, USA). For the lentivirus packaging, HEK293T cells (ATCC, Manassas, VA, USA) were co-transfected with a control vector or reconstructed plasmid and lentiviral packaging plasmid mix using X-tremeGENE HP DNA Transfection Reagent (Roche, Switzerland). At 48 h after transfection, the virus-containing medium was collected and filtered through a 0.45 μm filter and then added to NCIH1299 cells (ATCC, Manassas, VA, USA). Finally, the genetically modified target cells were selected by 2 μg/mL puromycin after 48 h of infection. The real-time quantitative PCR (RT-qPCR) assay was applied to confirm whether P53 expression was successfully induced in DOX-treated target cells.

### 4.4. Real-Time Quantitative PCR

Total RNA was isolated from cells using the TRIzol reagent (Sigma Aldrich, Burlington, MA, USA). Then, 3 μg of total RNA per sample was reverse-transcribed using the GoScript Reverse Transcription System Kit (Promega, Madison, WI, USA) following the manufacturer’s protocol. RT-qPCR was performed with FastStart™ Universal SYBR^®^ Green Master (ROX) (Roche, Switzerland) from Applied Biosystems (ThermoFisher Scientific, Waltham, MA, USA) in a 20 μL reaction volume according to the manufacturer’s protocols. Relative mRNA levels of genes were normalized to endogenous β-actin expression using the comparative Ct method. The primers used for RT-qPCR are listed in Appendix A.

### 4.5. Western Blot Analysis

To determine the P53-associated protein expression after drug treatment, cells were incubated in 6-well plates and 0.5 μg/mL DOX was added to stimulate the P53 expression. After treatment with indicated conditions for 24 h, the whole cell lysate proteins were collected and analyzed by Western blotting with the corresponding specific primary antibodies and an appropriate HRP-conjugated secondary antibody, followed by HRP activity-based signal detection. The following primary antibodies were used: CDKN1A (sc-6246) and P53 (sc-126) (Santa Cruz Biotechnology, Santa Cruz, CA, USA); MDM2 (ab259265) and PUMA (ab9643) (Abcam, Waltham, MA, USA); and P53 (Ab-3). HRP-conjugated anti-rabbit IgG was purchased from Cell Signaling Technology (Danvers, MA, USA) and anti-mouse IgG was purchased from BBI (Shanghai Sangon Biotech, Shanghai, China). Proteins were detected using the Western Lightning Plus-ECL Reagent (PerkinElmer, Boston, MA, USA) and the densitometry analysis was performed using ImageJ software.

### 4.6. Immunofluorescence

Cells were fixed in 4% paraformaldehyde (PFA) in phosphate-buffered saline (PBS) for 15 min, incubated in 0.2% Triton X-100 for 5 min and in PBS with 1% BSA for 10 min, and stained overnight with primary antibody at 4 °C (wtP53 protein-specific antibody PAb 1620 and mutP53 antibody PAb 240, Merck Millipore, Boston, MA, USA). Then, separate staining with fluorescein-labeled secondary antibody (Alexa Fluor 555 for PAb 1620 and Alexa Fluor 488 for PAb 240, Invitrogen, Waltham, MA, USA) was performed for 1 h at room temperature, followed by three washes and staining with DAPI. The cells were finally washed and visualized under a fluorescence confocal microscope (Carl Zeiss, Oberkochen, Germany).

### 4.7. Cell Viability Assay

The cells were plated in a 96-well plate at a density of 1 × 10^3^ cells per well for a 120 h treatment and at a density of 3 × 10^3^ cells per well for a 72 h treatment to analyze cell viability. In this study, cell viability tests were used twice. One was to compare the effect of wild-type or mutant P53 expression on the viability of NCIH1299 cells, in which each cell line was treated with or without DOX for 120 h. The other was to compare the effects of COTI-2 or ATO on the viability of cells expressing different mutant P53, in which each cell line was treated with COTI-2 and ATO for 72 h at the final concentration gradients, respectively.

After treatment, 100 μL of alarmarBlue solution (Invitrogen, Carlsbad, California, USA) was added to each well and the plates were incubated at 37 °C for an appropriate duration. The change in the fluorescence of the test reagent was measured using the excitation or emission wavelengths set at 530/590 nm. The cell growth inhibition rate was calculated as follows:Cell Growth Inhibition (%) = (OD590-T- OD590-BLK)/(OD590-T0- OD590-BLK) × 100%(1)

OD590-T was the absorbance value of each well with a specific compound concentration, OD590-T0 was the absorbance value of well without compound, and OD590-BLK was the background value.

### 4.8. Luciferase Reporter Gene Assay

The luciferase reporter gene plasmids pGL4.10-PUMA-Luc, pGL4.10-CDKN1A-Luc, pGL4.10-BAX-Luc, and pGL4.10-MDM2-Luc were constructed in this study. The methods are as previously described before [45,46]. The binding sequences of these promoters of P53 target genes are shown in Appendix A. The pcDNA3.1-GFP plasmid, which constitutively expresses green fluorescent protein, was acquired from Chipscreen Biosciences Co., Ltd. and was used for transfection normalization. Cells were seeded in 180 μL of the medium at 40% density in 96-well plates for 24 h. For each well, 100 ng of plasmid (40 ng of P53-expressing plasmid, 40 ng of luciferase reporter plasmid, and 20 ng of GFP plasmid) and 10 μL Opti-MEM (ThermoFisher Scientific, Waltham, MA, USA) were pre-mixed and left to stand for 5 min. Then, 0.15 μL X-tremeGENE HP DNA Transfection Reagent was added for each well and thoroughly mixed. After further incubation for 15 min at room temperature, the mixture was added to the medium. After incubation at 37 °C for 24 h, cells were treated with different final concentrations of compounds. After another 24 h, the cells were lysed and the luciferase signal was determined using a luciferase assay kit (Promega, Madison, WI, USA) according to the manufacturer’s instructions. The luminescent signal was normalized against that of GFP fluorescence for each sample.

### 4.9. The Response Modes of P53 Dominant Negative Effects Predicted by Mathematical Modeling

When the ratio of wild and mutant P53 monomers (p, q, and p + q = 1) in the system was determined, the proportion of the theoretical formation of P53 dimers (wt/wt, wt/mut, or mut/mut) could be calculated (p^2^, C(2,1)*p*q, q^2^) and the possibility of these monomers combined into different tetramers could be derived (4 wt: p^4^, 3 wt/1 mut: C(4,1)*p^3^q, 2 wt/2 mut: C(4,2)*p^2^q^2^, 1 wt/3 mut: C(4,1)*pq^3^, 4 mut: q^4^). There were 4 categories (wt = 4, wt ≥ 3, wt ≥ 2, and wt ≥ 1) that P53 tetramers could be divided into, and the probability of their mathematical distribution can be obtained by the proportion of the tetramers.

### 4.10. Data Analysis and Statistical Methods

T Each qPCR and Reporter Gene experiment was performed in triplicates. All data were expressed as mean ± standard deviation (x ± s). *p* < 0.05 was used as the criterion for statistical significance. The t-tests were used for two independent samples with homogeneity of variance and Welch-corrected t-tests were used for those with heterogeneity of variance. The two-stage step-up method of Benjamini, Krieger, and Yekutieli was used for multiple t comparisons. Data were analyzed using the GraphPad Prism version 8 (GraphPad, La Jolla, CA, United States).

## Figures and Tables

**Figure 1 ijms-23-13867-f001:**
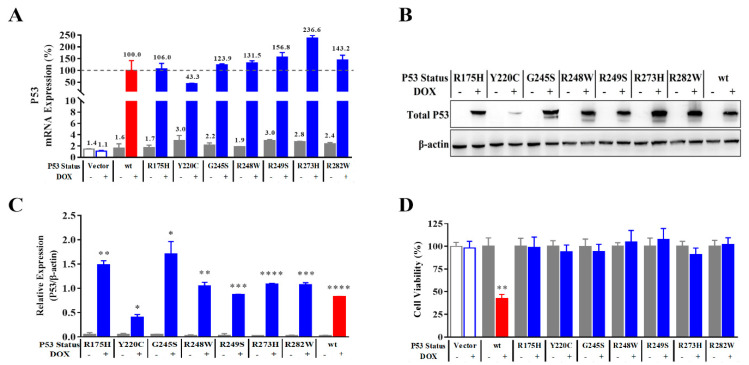
DOX-induced expression of wtP53 or mutP53. (**A**) RT-qPCR analysis of the expression of mRNA showed wtP53 or mutP53 was significantly induced by 0.5 μg/mL of DOX for 24 h in the cell lines infected with lentivirus containing the coding sequence (*n* = 3). (**B**) Band diagrams of total P53 protein in the stable transgenic NCIH1299 cell lines (treated without or with DOX) were determined by Western blot analysis. (**C**) The expression of P53 in the stable transgenic NCIH1299 cell lines was determined by Western blot analysis. (**D**) The cell viability alterations after inducing different statuses of P53 protein for 120 h (*n* = 6). Data are presented as the mean ± S.D. obtained from three separate experiments. Compared to the corresponding non-DOX group, * *p* < 0.05, ** *p* < 0.01, *** *p* < 0.001, and **** *p* < 0.0001.

**Figure 2 ijms-23-13867-f002:**
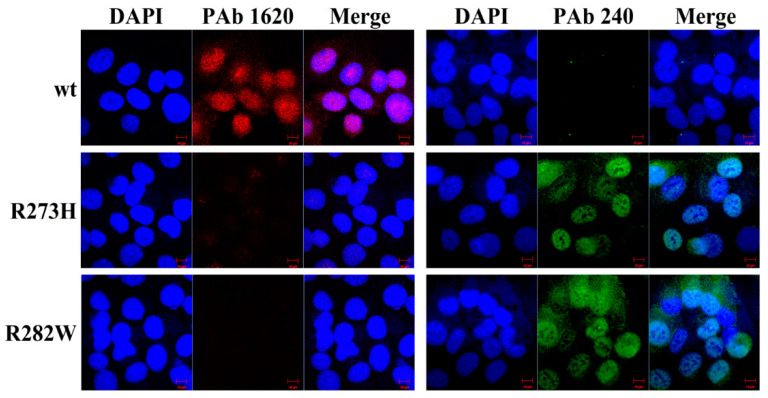
Immunofluorescence results of the different statuses of P53 (wtP53 antibody PAb 1620 stained with fluorescein-labeled secondary antibody (red, Alexa Fluor 555); mutP53 antibody PAb 240 stained with fluorescein-labeled secondary antibody (green, Alexa Fluor 488)). DAPI observed the cell nucleus. Representative scopes are shown (*n* = 3). The objective lens is 100 × magnifications and the scale bar represents 20 μm.

**Figure 3 ijms-23-13867-f003:**
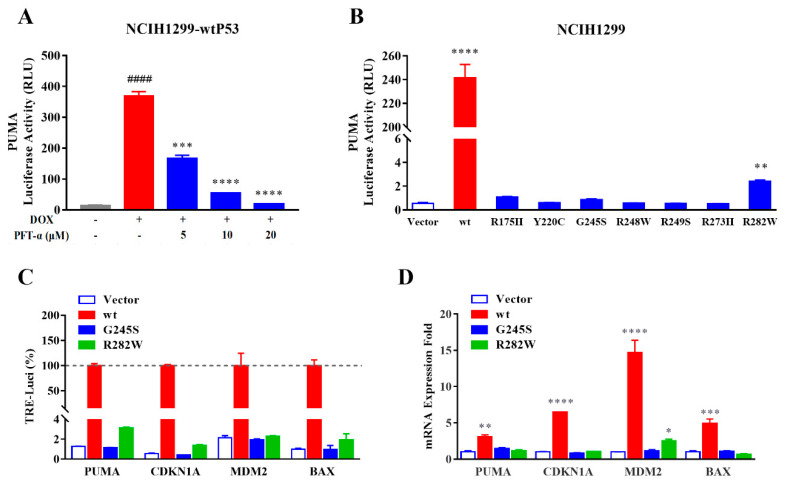
Activity status of P53 correctly reflected by improved reporter gene system. (**A**) Activation of the TRE of PUMA by wtP53 (red), P53 inhibitor PFT-α reduced the reporter gene signal in a dose-dependent manner (blue) (*n* = 3). (**B**) Induction of PUMA reporter gene by wtP53 (red) and mutP53 (blue) (*n* = 3). (**C**) Using reporter gene with P53-TRE of four target genes (*PUMA*, *CDKN1A*, *MDM2*, and *BAX*) to measure the effect of different P53 statuses (*n* = 3). (**D**) RT-qPCR assay of the four target genes in corresponding samples of (**C**) (*n* = 3). Data are presented as the mean ± S.D. obtained from three separate experiments. Compared to the non-DOX group in (**A**), ^####^
*p* < 0.0001; compared to the DOX but non-PFT-α group in (**A**), *** *p* < 0.001 and **** *p* < 0.0001; compared to the corresponding vector group, * *p* < 0.05, ** *p* < 0.01, *** *p* < 0.001, and **** *p* < 0.0001.

**Figure 4 ijms-23-13867-f004:**
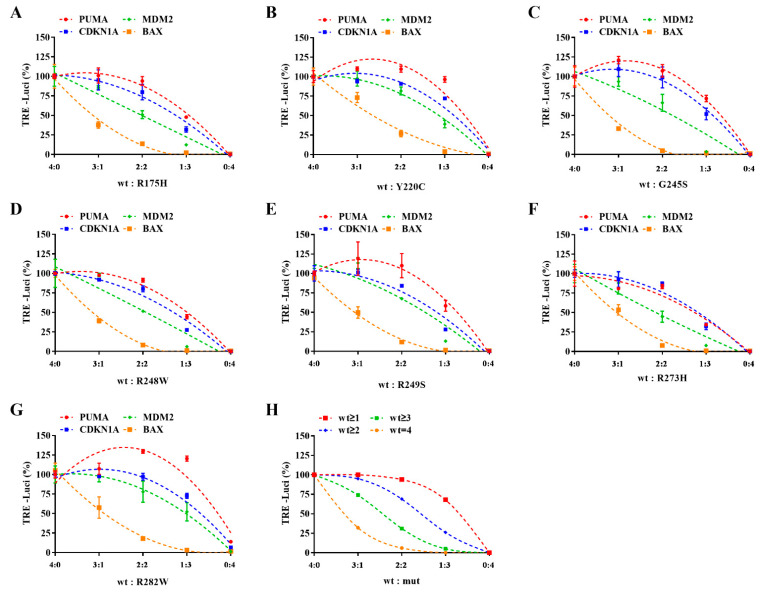
The response modes of the TRE of P53 target genes to DNE were observed by reporter gene assay and predicted by mathematical modeling. (**A**–**G**) The wtP53 and mutP53 expression plasmids were accurately pre-mixed according to the indicated proportion and transiently co-transfected with different reporter gene plasmids into NCIH1299 cells. After DOX treatment, the reaction of reporter genes to the mixed expression was observed and recorded (*n* = 3). Data are presented as the mean ± S.D. obtained from three separate experiments. (**H**) Four TRE response modes were deduced from the given wt/mutP53 monomer proportion.

**Figure 5 ijms-23-13867-f005:**
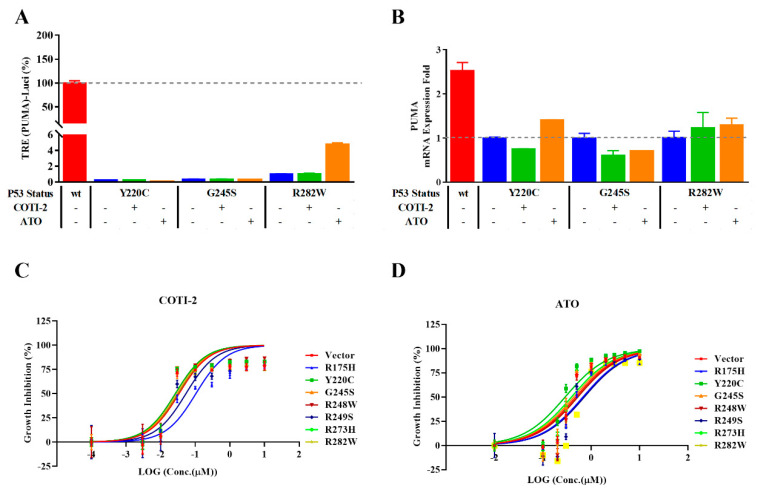
An improved reporter gene system was applied for evaluating mutP53 reactivators. (**A**) Reporter gene assay for measuring TRE activation of *PUMA* (*n* = 3). (**B**) RT−qPCR analysis of *PUMA* mRNA expression after treatment with P53 activators (0.1 μM COTI−2 or 1 μM ATO) for 24 h. (**C**,**D**) Cell growth inhibition of different P53 status and negative control after treatment with COTI−2 at the final concentrations of 0, 0.003, 0.01, 0.03, 0.1, 0.3, 1, 3, and 10 μM or ATO at the final concentrations of 0, 0.1, 0.2, 0.3, 0.5, 1, 2, 3, 5, and 10 μM, respectively (*n* = 3). Data are presented as the mean ± S.D. obtained from three separate experiments.

**Figure 6 ijms-23-13867-f006:**
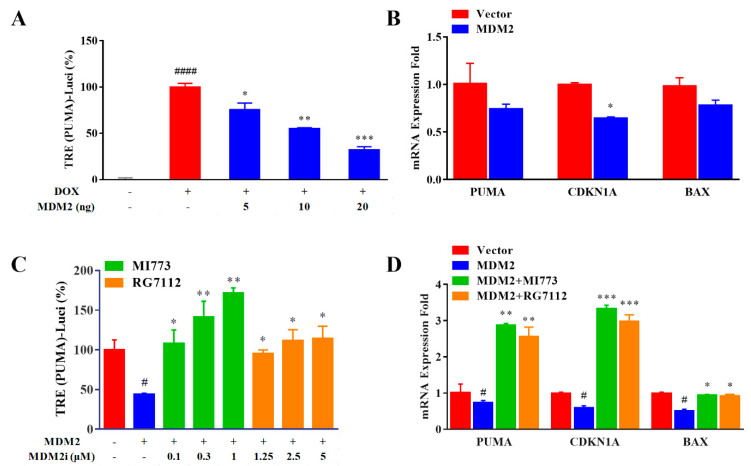
Improved reporter gene system was applied for evaluating MDM2 inhibitors. (**A**) The transfected overexpression of MDM2 decreased the reporter gene signal of *PUMA* in a dose-dependent manner (*n* = 3). (**B**) RT−qPCR analysis to verify the influence of transfected MDM2 plasmid (20 ng per well) on reporter gene system (*n* = 3). (**C**) MDM2 inhibitors (MI773 and RG7112) were tested by the *PUMA* reporter gene in MDM2−overexpressed NCIH1299−wtP53 (*n* = 3). (**D**) RT-qPCR analysis to verify the influence of MDM2 inhibitors (1 μM MI773 or 5 μM RG7112) on the transcriptional level of P53 target genes (*n* = 3). Data are presented as the mean ± S.D. obtained from three separate experiments. Compared to the non-DOX group in (A), ^####^
*p* < 0.0001; compared to the DOX but non-*MDM2* group in (**A**), * *p* < 0.05, ** *p* < 0.01, and *** *p* < 0.001; compared to the corresponding vector group in (**B**), * *p* < 0.05; compared to the non-MDM2 group in (**C**,**D**), ^#^
*p* < 0.05; compared to the *MDM2* group in (**C**,**D**), * *p* < 0.05, ** *p* < 0.01, and *** *p* < 0.001.

**Figure 7 ijms-23-13867-f007:**
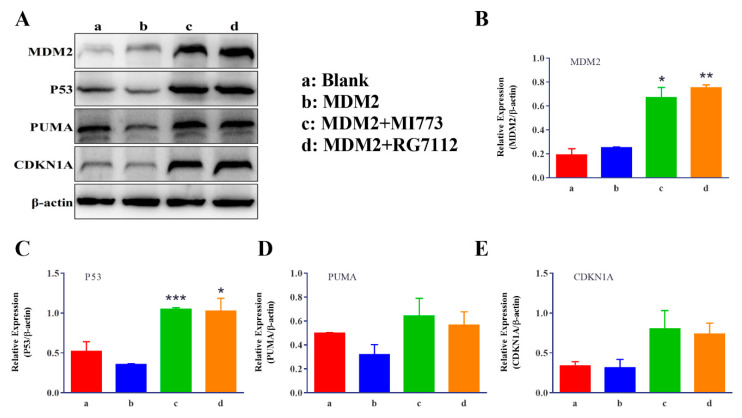
(**A**) Band diagrams of proteins associated with P53 signal pathways with the experimental condition of 1 μM MI773 or 5 μM RG7112 determined by Western blot analysis, the a−d lanes were blank, MDM2 plasmid, MDM2 plasmid and 1 μM MI773, MDM2 plasmid and 5 μM RG7112, separately. (**B**) The expression of MDM2 with the experimental condition of 1 μM MI773 or 5 μM RG7112 was determined by Western blot analysis. (**C**) The expression of P53 with the experimental condition of 1 μM MI773 or 5 μM RG7112 was determined by Western blot analysis. (**D**) The expression of PUMA with the experimental condition of 1 μM MI773 or 5 μM RG7112 was determined by Western blot analysis. (**E**) The expression of CDKN1A with the experimental condition of 1 μM MI773 or 5 μM RG7112 was determined by Western blot analysis (*n* = 3). Compared to the MDM2 group, * *p* < 0.05, ** *p* < 0.01, *** *p* < 0.001.

**Figure 8 ijms-23-13867-f008:**
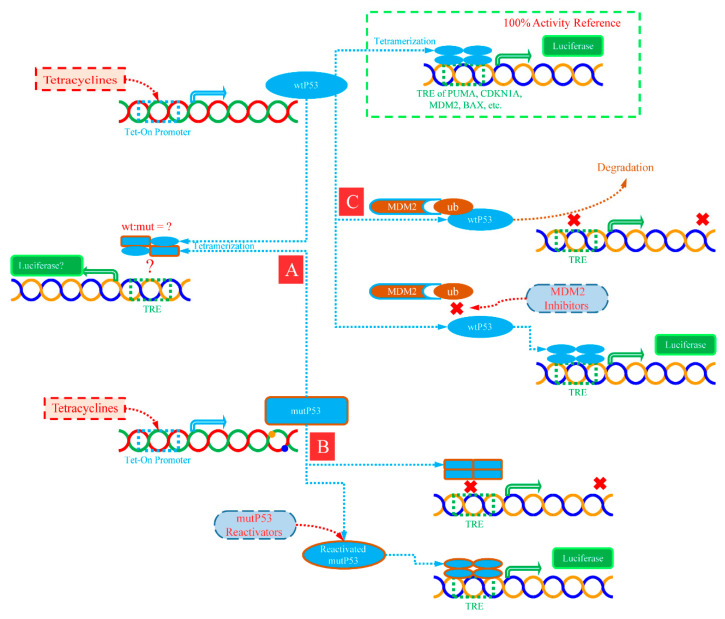
Schematic overview of the improved reporter gene assay and its application. The strategy of tetracycline-inducible expression made P53 expressed controllable only during the experiment. Thus, it provides a 100% active reference with the whole activation state of wild-type P53 for the reporter gene system. The method of calculating the percentage relative to the 100% reference can more objectively reflect the actual biological activity. Based on this solution, incomplete same response modes of typical transcriptional response elements to the dominant negative P53 mutation effect can be observed (**A**). The system can also be applied to evaluate the ability of compounds to restore P53 activity, including but not limited to mutant P53 reactivators (**B**) and MDM2 inhibitors (**C**).

**Table 1 ijms-23-13867-t001:** Deduction of the probability distribution of each homo- or hetero-dimer or -tetramer and the activation percentage in each TRE response mode based on the given wt/mutP53 monomer proportion.

Proportion		Deduced Probability Distribution								Deduced Activation Percentage			
Monomer		Dimer			Tetramer					TRE Response Mode			
**wt**	**mut**	wt/wt	wt/mut	mut/mut	4 wt	3 wt/1 mut	2 wt/2 mut	1 wt/3 mut	4 mut	wt = 4	wt ≥ 3	wt ≥ 2	wt ≥ 1
**4**	**0**	100%	0%	0%	100%	0%	0%	0%	0%	100%	100%	100%	100%
**3**	**1**	56%	38%	6%	32%	42%	21%	5%	0%	32%	74%	95%	100%
**2**	**2**	25%	50%	25%	6%	25%	38%	25%	6%	6%	31%	69%	94%
**1**	**3**	6%	38%	56%	0%	5%	21%	42%	32%	0%	5%	26%	68%
**0**	**4**	0%	0%	100%	0%	0%	0%	0%	100%	0%	0%	0%	0%

## Data Availability

Not applicable.

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
