# Peer review of "Improving Reporter Gene Assay Methodology for Evaluating the Ability of Compounds to Restore P53 Activity"

_ijms, 2022, doi:10.3390/ijms232213867_

Round 1

Reviewer 1 Report

In this manuscript, the authors developed the system to measure p53’s transcriptional activity without affecting the cytotoxicity and demonstrated that the current system is potentially useful to develop mutant p53 reactivators. Tet on system is quite old fashioned, and they applied the current system to the recently developed anti-cancer drugs. Although the authors claimed that the reporter system is very sensitive, Figure 6C and D showed that quantitative RT-PCR is more sensitive than reporter system..

Figure 4. The response model of p53 is somewhat difficult to understand. If the authors add the simple diagram to explain this model, the data will be easily understood. In addition, the comparison between the experimental data and deduced model should be discussed.

Figure 5A and B. The authors showed the results of three mutants. The authors should include the results of R175H, R248W, R249S and R273H, because the p53 reactivator may contribute to other mutants and Figure 5C and D contain the data of other mutants.

Figure 6A and B. The authors should show the expression of p53 protein to confirm the degradation of p53 by MDM2.

Figure 7A. The authors should include Western blot data of Bax protein. In Figure 6, the authors demonstrated that the expression of Bax was not changed by Mdm2 inhibitor.

Reviewer 2 Report

The authors studied the use of tetracycline inducible expression of wild-type P53 for evaluating the 100% activity reference, and presenting a calculation model for future drug development in the inactivation of tumors.

This article is generally interesting with a good  description of the objectives and careful design of experimental works, as well as  a solid discussion of experimental results. Based on these, I recommended the publication of this article in IJMS journal as it is. 

Author Response

Response: Thanks for your review. We appreciate the time and effort that you dedicated to providing feedback on our manuscript.

Reviewer 3 Report

In their article, Han and colleagues describe a reporter gene assay for p53 activity, with the aim to generate a system that would allow studying p53 activity and therapeutic strategies to restore p53 function in tumor cells. For their purpose, they use the NCIH1299 non-small cell lung cancer cell line that does not express p53 protein, restored with Tetracycline-inducible wild type or different p53 dominant negative mutant versions. A reporter gene, i.e. luciferase driven by the p53 TRE of p53-response genes PUMA, BAX, CDKN1A and MDM2, reports p53 activity. The reporter system was validated with the addition of PFT-a, a small molecule p53 inhibitor. Next, Han et al use their system to test expression of the 4 above p53-response genes when wild type p53 is expressed along with different p53 mutant versions. This is an interesting aspect of the work as their results suggest that wild type:mut p53 tetramers do affect the 4 p53-response genes differently. They also use two different small molecule MDM2 inhibitors to evaluate their system, and introduce an algorithm for the evaluation of compounds that restore p53 activity.

Overall, the work introduces an interesting combined experimental and mathematical modeling system for the evaluation of p53 activity. Scientifically, the authors results point towards different biological outputs of dominant negative p53 mutant versions on transcriptional activity.

Major points:

Related to the interesting finding that wild type:mut p53 tetramers do affect p53-response genes differently:

In Figure 2, the authors show that the R273H and the R282W mutant p53 versions are located in the cytoplasm. Does mutant p53 translocate to the nucleus when complexed with wild type p53 in their system? Is there differences between the mutants?

What are the determinants of the varying transcriptional response? Can the authors experimentally address this question?

Information on detailed description of the mathematical modeling is missing in the Materials and Methods section. It should be included by the authors.

Round 2

Reviewer 1 Report

The authors performed the extensive experiments to respond to the reviewers' comment. And now the manuscript is suitable for the publication.  

Author Response

Thanks for your review. We appreciate the time and effort that you dedicated to providing feedback on our manuscript.

Reviewer 3 Report

The authors have not attempted to experimentally address the points 1 and 2 raised by this reviewer, and there is no satisfactory explanation why.
